# Simulation and Economic Analysis of the Biotechnological Potential of Biomass Production from a Microalgal Consortium

**DOI:** 10.3390/md21060321

**Published:** 2023-05-26

**Authors:** Christian Ariel Cabrera-Capetillo, Omar Surisadai Castillo-Baltazar, Moisés Abraham Petriz-Prieto, Adriana Guzmán-López, Esveidi Montserrat Valdovinos-García, Micael Gerardo Bravo-Sánchez

**Affiliations:** 1Departamento de Posgrado, Doctorado en Ciencias de la Ingeniería, Tecnológico Nacional de México en Celaya, Antonio García Cubas #600 Pte., Colonia Alfredo V. Bonfil, Apartado Postal 57, Celaya 38010, Guanajuato, Mexico; d2203012@itcelaya.edu.mx; 2Programa de Biotecnología, Universidad de Guanajuato, Mutualismo #303, Colonia La Suiza, Celaya 38060, Guanajuato, Mexico; omar.castillo@ugto.mx; 3División Académica Multidisciplinaria de Jalpa de Méndez (DAMJM), Universidad Juárez Autónoma de Tabasco (UJAT), Carret. Estatal Libre Villahermosa-Comalcalco Km. 27+000 s/n Ranchería Ribera Alta, Jalpa de Mendez C.P. 86205, Tabasco, Mexico; moises.petriz@ujat.mx; 4Departamento de Ingeniería Química, Tecnológico Nacional de México en Celaya, Antonio García Cubas #600 Pte., Colonia Alfredo V. Bonfil, Apartado Postal 57, Celaya 38010, Guanajuato, Mexico; adriana.guzman@itcelaya.edu.mx; 5Departamento de Ingeniería Bioquímica, Tecnológico Nacional de México en Celaya, Antonio García Cubas #600 Pte., Colonia Alfredo V. Bonfil, Apartado Postal 57, Celaya 38010, Guanajuato, Mexico

**Keywords:** pigment, chlorophyll, microalgae consortium, techno-economic, bioeconomic, cost

## Abstract

The biomass of microalgae and the compounds that can be obtained from their processing are of great interest for various economic sectors. Chlorophyll from green microalgae has biotechnological applications of great potential in different industrial areas such as food, animal feed, pharmaceuticals, cosmetics, and agriculture. In this paper, the experimental, technical and economic performance of biomass production from a microalgal consortium (*Scenedesmus* sp., *Chlorella* sp., *Schroderia* sp., *Spirulina* sp., *Pediastrum* sp., and *Chlamydomonas* sp.) was investigated in three cultivation systems (phototrophic, heterotrophic and mixotrophic) in combination with the extraction of chlorophyll (*a* and *b*) on a large scale using simulation; 1 ha was established as the area for cultivation. In the laboratory-scale experimental stage, biomass and chlorophyll concentrations were determined for 12 days. In the simulation stage, two retention times in the photobioreactor were considered, which generated six case studies for the culture stage. Subsequently, a simulation proposal for the chlorophyll extraction process was evaluated. The highest microalgae biomass concentration was 2.06 g/L in heterotrophic culture, followed by mixotrophic (1.98 g/L). Phototrophic and mixotrophic cultures showed the highest chlorophyll concentrations of 20.5 µg/mL and 13.5 µg/mL, respectively. The simulation shows that higher biomass and chlorophyll production is attained when using the mixotrophic culture with 72 h of retention that we considered to evaluate chlorophyll production (*a* and *b*). The operating cost of the entire process is very high; the cultivation stage has the highest operating cost (78%), mainly due to the high energy consumption of the photobioreactors.

## 1. Introduction

Microalgae have been widely studied since the mid-20th century, due to their ability to produce primary and secondary metabolites and their role in important industrial products (pigments, biofuels, biofertilizers and food supplements) [1,2]. Microalgae are of great importance in several areas of study due to their high biotechnological potential. In recent decades, microalgae have been used for the production of products in the areas of food, biofuels, animal food and pharmaceuticals [3,4]. The increase in demand for food, energy and materials has led to a fundamental role for microalgae as a raw material in the bioeconomy [5].

Current research trends for microalgae biomass have migrated from interest in biofuels to food supplements, sustainable production of microalgal biomass, biofertilizers, and bioremediation of water, soil and air [1,6]. The products that can be obtained from it have been firmly positioned among health foods known as “superfoods”, which are “rich in bioactive compounds” [7,8]. Microalgal biomass and its components, such as lipids, proteins, chlorophyll and amino acids, have acquired economic importance at the industrial level [9]. However, the cost of products derived from microalgae is still high; therefore, reducing production costs is a focal point of research. Currently, several production systems have been implemented to reduce costs, and these are validated using techno-economic analyses [10].

Microalgae are some of the main photosynthesizers of pigments at the industrial level, which gives them a high value in the world market of commercial pigments [11]. The most commonly used are blue-green microalgae (BGA) containing chlorophyll *a* and chlorophyll *b* [12]. The bioactive compounds with health benefits that can be obtained from microalgae include pigments such as carotenoids, chlorophyll, and phycobiliproteins, which have been studied for their anticancer, antioxidant, and antihypertensive properties [8]. Pigments are considered compounds of high value in the food sector because they are useful as food supplement promoters [13,14], due to their high nutritional value and competitive production cost [11].

Microalgae-based functional foods have not been exploited due to the high cost associated with microalgae harvesting, maintenance and extraction of their compounds of interest [8]; this cost is incurred because specific nutrients and equipment are required. For the production of microalgal biomass, alternatives must be sought in the cultivation system in order to reduce operating, input and maintenance costs [15]. The main challenges in cost reduction are correct selection of microalgae species, optimization of cultivation conditions, economically viable harvesting and high input costs [16]. To obtain high concentrations of biomass, heterotrophic and mixotrophic growth conditions are used, providing a source of organic carbon, in addition to synthetic mineral media that favor growth [17].

Microalgae can be grown under autotrophic, phototrophic, heterotrophic and mixotrophic culture conditions [18]. The most studied and conventional condition is phototrophic growth, which is used because it is relatively inexpensive; this is because it does not require a source of organic carbon, and with aeration, the microalgae obtain the carbon necessary for their proliferation and for the generation of pigments [13,19]. The growth conditions depend on the energy source (light) and the type of carbon needed (organic and inorganic) [13]. For phototrophic culture, a source of light energy (natural or synthetic) is necessary for the microalgae to photosynthesize; they must also have a source of inorganic carbon. In most cases, bubbling aeration is used, adding CO_2_ to the culture [20]. For heterotrophic culture, an organic energy source and an organic carbon source are required; in most investigations, glucose is used because it generates the least energy expenditure for the cells, and it is necessary that the culture is in total darkness [18]. For mixotrophic culture, the energy source must be a combination of organic and inorganic, and the carbon source must be organic and inorganic [18].

It has been reported that one of the advantages of using a heterotrophic culture is that it has a greater accumulation of lipids in the biomass, a high cell density, and the equipment design is simple, compared to phototrophic culture [20] (a type of culture highly used for the generation of biofuels). Regarding mixotrophic culture, it has been reported that a high cell density, prolonged growth, and a high concentration of pigments and proteins are achieved; these qualities are mainly utilized in the health sector [21,22].

The use of consortia of microalgae has been investigated in different fields of study. These consortia can be natural or artificial; natural consortia are found in combination with bacteria, while artificial consortia are composed of microalgae or microalgae with bacteria [23,24]. Natural consortia coexist, having a cooperative interaction between the microorganisms present [23]. Among the applications of these natural consortia are the bioremediation of soils with the use of biofertilizers, of air with the mitigation of atmospheric carbon, and of water with the removal of nitrogen, phosphorus, metallic contaminants, and organic load from all types of wastewater (industrial, dairy, domestic, etc.) [25,26,27]. At the end of the remediation process, the products obtained are the resulting treated water (primary, secondary and tertiary treatment) and the biomass of microalgae that can be used to obtain other compounds of interest [28]. It is essential that, according to the final use of the microalgal biomass production, an appropriate culture mode be selected so that the profitability of the process may be increased [18]. In addition, having a theoretical model of the production process facilitates techno-economic evaluation [4].

In this paper, the biotechnological potential of cultivating a natural consortium of regional microalgae to produce biomass and chlorophyll (*a* and *b*) under different growth conditions (phototrophic, heterotrophic and mixotrophic) in the cultivation stage is explored, based on experimental data and with the help of SuperPro Designer v10.0 software. A theoretical model was also developed as a design proposal for the microalgal biomass production, which allows technical and economic evaluation of the feasibility of the proposed process. Parameters such as energy consumption and operating costs were primarily evaluated.

## 2. Results and Discussions

### 2.1. Experimental Biomass and Chlorophyll Production

The production of microalgal biomass and total chlorophyll in different growth types is shown in Figure 1 and Figure 2, respectively. The maximum biomass values obtained were 0.5740 g/L for phototrophic, 1.9888 g/L in mixotrophic and 2.0687 g/L in heterotrophic culture. The increase in biomass production in heterotrophic and mixotrophic cultures, corresponding to more than three times that in the phototrophic culture, can be attributed to the capacity of all the microalgae present in the consortium to take advantage of glucose as a source of organic carbon, presenting a total soluble organic carbon remobilization of 90–95% in both cases. In addition, the environmental CO_2_ may not have been sufficient to favor growth in the phototrophic cultures. The total chlorophyll concentration (chlorophyll *a* and *b*) is related to photosynthetic activity. The heterotrophic culture was the one that presented the lowest concentration of chlorophyll, 3.3564 µg/mL, since in this culture, photosynthetic activity was limited by placing it in total darkness. The phototrophic and mixotrophic cultures showed the highest concentrations, 20.5450 µg/mL and 13.5415 µg/mL, respectively.

The use of cultures of a single species of microalgae is the most common, since all the optimal growth conditions are known in detail; the most used are *Dunaliella*, *Spirulina* and *Chlorella*, of which the production costs are USD3–4/kg, and the sales costs are USD20–44/kg dry biomass; these costs are dependent on the purity of the cultures and the system in which they are cultivated [29]. Cultures with microalgae consortia are not very well studied, because it is not known with certainty how the population’s interaction will react to the established conditions; therefore, it is important to analyze all possible conditions. Naturally, in water bodies that harbor microalgae, they are always found in combination with other species of microalgae, and also with fungi, bacteria and/or protozoa in general. It is important to find a product in which these consortia can be exploited, with chlorophyll being one of the main products of interest. The amount of chlorophyll *a* and *b* in pure microalgae cultures is around 0.5 and 1% of the dry biomass [30].

Taking into account the experimental results, retention times of 72 h and 288 h were considered as the basis for comparing the different types of cultivation with the help of the simulator, since it was observed that the highest biomass and chlorophyll concentrations were obtained at these points.

### 2.2. Results of the Techno-Economic Evaluation for the Production of Biomass from a Microalgal Consortium

Figure 3 shows the flowsheet developed for the simulation of the cultivation stage, specifically for the M1 case (the conditions for this case are described in Section 3.5 and in Table 4), in which a biomass concentration of 1.57 g/L is achieved at the bioreactor outlet. In a similar way, another five study cases were developed in the simulation in SuperPro Designer, with which the six proposed cases were evaluated. According to the results of the simulation, 2177 units of 300 L are needed to cover 1 ha of the surface to be used as the cultivation stage, considering a feeding flow of 8160 L/h for the cases that consider 72 h as the residence time (cases F1, H1 and M1), and a feeding flow of 2040 L/h for the cases that consider 288 h (F2, H2 and M2). The annual biomass production estimated with the help of the simulator is shown in Table 1 and Figure 4 for all the study cases. The highest annual biomass production corresponds to the heterotrophic culture (H1); however, it is not the culture with the highest chlorophyll production, due to the low concentration obtained. The case of the mixotrophic culture (M1) has the second highest biomass production of all the cases, and even though the biomass concentration is not the highest, the chlorophyll production is higher than in the other cases.

In continuous operation mode and with the help of the simulator, the evaluation of the operating cost indicates that when the cultures are operated at 72 h residence time, the cost is only 0.5% higher. This is due to the increase in raw material consumption, since the other costs that add to the operating cost, such as labor, facility-dependent costs and utilities, remain the same in all cases. Case H1 has the highest raw material consumption cost (62,019 USD/year), followed by case M1 (45,874 USD/year), while for cases F2, H2 and M2, the raw material consumption cost is the lowest and similar for all cases (15,119 USD/kg to 15,291 USD/year). The labor cost is low and the same for all cases (8327 USD/year), since it was considered a small expense for operators (0.39 USD/h). Similarly, the dependent cost of the facilities is similar for all cases (4,768,000 to 4,770,000 USD/year); there is only a difference if the culture time is 72 h or 288 h. This cost is related to the payment of equipment maintenance, taxes, and others. The same is true of the costs associated with services (2,701,000 to 2,702,000 USD/year), being slightly lower for the cases with a 72 h cultivation time. Energy consumption is very high, raising the cost of operation to a great extent; therefore, other alternatives for cultivation should be proposed and evaluated in order to reduce this energy consumption or increase biomass production.

### 2.3. Results of the Techno-Economic Evaluation of the Production of Chlorophyll from a Microalgal Consortium

For the design of the chlorophyll production process, the case of the cultivation stage in a mixotrophic culture system (M1) was considered, since according to the simulation results, it is the case with the highest chlorophyll production. The process flow diagram for chlorophyll production that was developed in the simulator is shown in Figure 5.

The operating cost of the process is the sum of the cost of raw materials (21.8%), labor (0.4%), facility-dependent costs (20.7%) and service costs (57%), as seen in Figure 6a. The cultivation stage represents 78% of the operating costs of the process (Figure 6b), which are distributed as follows: (a) 1.68% is due to the consumption of raw materials, of which 32.46% is due to the phosphorus source (K_2_HPO_4_ and KH_2_PO_4_), 26% to fresh water, 23% to the carbon source (glucose), and 7.8% to the nitrogen source (EDTA), and the other nutrients make up the rest. The consumption of fresh water in this process is high, since the microalgae need an aqueous medium for their proliferation, which is why the combination of these processes with wastewater treatment (or other water treatments) is sought to reduce the impact of this factor [31,32]. (b) Labor accounts for 0.23% of the operating cost of the process; for this study, only operators were considered, at an annual cost of USD8327. Only operators were considered able to maintain the operation of the technologies involved in the process, at a cost of 0.37 USD/h. This variable may have a significant impact if the labor cost is higher. (c) The dependent cost of the facilities represents 25.14%; this cost is calculated as the sum of the costs associated with equipment maintenance, depreciation of the fixed capital cost, and some other expenses such as insurance, local taxes (property taxes) and others. All of these are determined as a percentage of the direct fixed capital cost, which in turn includes the sum of costs that are estimated as percentages of the total equipment purchase cost. The cultivation stage occupies 95% of the equipment purchase cost for the process (Table 2), so the dependent cost of the facilities is also high at this stage. (d) Finally, 72.94% of the operating cost of the cultivation stage is due to the services necessary for the process, such as standard power and chilled water. Closed cultivation systems are highly energy-consuming [33]; in this case, the column photobioreactors considered for the process design consume 173.21 kWh/kg of the biomass produced, and represent 99.95% of the total consumption of the process. In this calculation, the consumption required to maintain artificial lighting in the cultures that need it was not considered, so the consumption could still be much higher. Energy consumption is the factor with the greatest impact on the operating costs of the process, since it is necessary to maintain good culture mixing. It is important to look for an alternative to reduce consumption; open system photobioreactors (e.g., thin-layer, raceways, lagoons) can be implemented, which in terms of design and operation are considerably less expensive than closed photobioreactors (e.g., flat-panel, column, tubular) [34].

The first harvesting stage represents 21% of the operating cost of the process, of which 93.6% is due to the consumption of the flocculant (chitosan and acetic acid). The flocculant (chitosan) was prepared at 1 g/L in a solution of 20% acetic acid and 80% water (*v*/*v*). At this concentration, the flocculant achieved flocculation and recovery of 98% of the biomass. We observed that the costs associated with the flocculant are higher compared to the costs of the nutrients needed for the culture; the costs associated with the flocculant represent 87.8% of the total cost of raw materials needed for the process. This is another factor with a high impact on the operating cost of the process, so it is necessary to seek to reduce this impact, either by evaluating the use of lower concentrations of the flocculant or by replacing it with one of lower cost and/or greater impact on the flocculation efficiency. However, the advantages that justify the use of chitosan in comparison with other organic and inorganic flocculants are its high biomass recovery, low contamination, no changes in the color of the microalgae, and no limitation of the recirculation of the culture medium [35].

In the second stage, the use of filtration technology was considered to concentrate the biomass to 200 g/L. This stage represents only 0.5% of the total operating costs of the process, of which 1.6% is due to the consumption of fresh water for washing the biomass, 0.8% is due to energy consumption, 6.1% is for labor, and 91.5% is associated with the costs associated with the facilities, such as equipment maintenance, among others. This technology has been compared in the literature with centrifugation (a widely used technique), and it has been shown that filtration has lower operating costs and lower energy consumption, saving 25–90% of energy [36,37].

Finally, in the extraction stage, the wet biomass is mixed with the solvent and then the chlorophyll is separated using evaporation. This stage consumes only 1% of the operating cost of the entire process. Some 8.7% of the operating cost of this stage is due to the purchase of the solvent (methanol) necessary for the extraction, 17.42% is due to labor, 16.8% is due to energy consumption (34 kWh/year), chilled water (10,565 MT/year), steam (122 MT/year) and glycol as cooling agent (2080 MT/year), and 57.1% is included in the costs dependent on the facilities, which are mainly due to the maintenance of the equipment. For this study, the cost of the equipment proposed by the simulator was considered for the primary harvesting, secondary harvesting and extraction stages, with the exception of the cultivation systems, whose individual price was set at USD1712. The cost of the equipment is listed in Table 2. The simulator has a database of design parameters which are used in the equipment model to obtain the equipment purchase cost, and the SuperPro Designer v10^®^ software uses the Chemical Engineering Cost Index to account for inflation to adjust the cost to different years (Intelligen, Inc., Scotch Plains, NJ, USA).

Production costs are high when the extraction of compounds of interest from microalgae biomass is intended, and often, the production costs in bioprocesses are not competitive when compared with processes for the alternative production of synthetic compounds [4]. One of the main factors that increases the cost of microalgal biomass production is the use of bioreactors, since they are more expensive due to their complex construction and high energy consumption [38]. On the other hand, the cost of production will also depend on the scale of production; Vázquez-Romero [1] estimates that microalgal biomass could be produced for 108.26 or 44 EUR /kg DW, and the cost will depend on the scale of production, with 1 to 100 ha covered by culture in photo-bioreactor. In our case, the cost of producing microalgae biomass under the photobioreactor cultivation scheme covering 1 ha of culture surface, and from a consortium of microalgae, was USD47/kg of wet biomass. It is difficult to make a comparison between the different processes and their production costs, since the evaluations are carried out considering different parameters, and these depend on the type of microalgae for biomass production, the production scale, operating conditions, and nutrient consumption, among many others factors that cause great diversity in techno-economic evaluations.

However, techno-economic analyses can be useful for assessing the commercial viability of any process, from microalgae culture to final product; they also help to establish the scale of any microalgae project/process, and to quantify the associated financial and technical risks. In this way, they are able to initiate an adequate strategy for the technical development of any process [38].

## 3. Materials and Methods

This study was divided into three stages. In the first stage, only the three types of biomass growth in the culture stage (phototrophic, heterotrophic and mixotrophic) were evaluated experimentally. Subsequently, considering the experimental results, in the second stage, the three types of growth were evaluated techno-economically with the help of a model created with the SuperPro Designer v10^®^ simulator (Intelligen, Inc., Scotch Plains, NJ, USA) for each case. In the third stage, a process model was developed as a proposal for the utilization of the biomass of the microalgae consortium and the extraction of total chlorophyll, thus evaluating both technically and economically the proposed process, and determining the factors with the greatest influence on operating costs.

### 3.1. Study Material

The microalgae consortium was collected from the Neutla dam, located in the community of Comonfort in the State of Guanajuato, Mexico (20°71′58.03.03″ N, 100°87′12.77″ W). The morphologically identified genera of microalgae in the natural consortium were *Scenedesmus* sp., *Chlorella* sp., *Schroderia* sp., *Spirulina* sp., *Pediastrum* sp., and *Chlamydomonas* sp. and bacteria. The consortium was initially adapted for 4 months to grow in BBM 3-N synthetic mineral medium under controlled conditions [35]. Figure 7 shows a microscope photo presenting the morphology of the type of microalgae mentioned above.

### 3.2. Determination of the Cell Growth

The biomass concentration of the microalgal strain was determined by measuring the optical density (OD) at a wavelength of 680 nm with a UV/Vis spectrophotometer [3,39]. It converted the OD680 values to dry cell weight (DCW) via a proper calibration between OD680 values and DCW. A strong linear relationship with R^2^ = 0.9792 between the OD680 and DCW is given as follows:DCW = 0.2983(OD680) − 0.05(1)

### 3.3. Determination of the Chlorophyll

The total chlorophyll (*a + b*) is evaluated by measuring the absorbance of the methanol extract at 652 nm and 665 nm [3,40]. The following equations were used to determine chlorophyll in (μg/mL) [40]:𝐶ℎ𝑙 𝑎 ≈ −8.0962𝑥𝐴652 + 16.5169𝑥𝐴665 (±0.04696 μg/mL)(2)
𝐶ℎ𝑙 𝑏 ≈ 27.4405𝑥𝐴652 − 12.1688𝑥𝐴665 (±0.05776 μg/mL)(3)

### 3.4. Microalgae Biomass Production

In the phototrophic and mixotrophic cultures, biomass production was carried out in an operating column photobioreactor fed with 1.0 L of BBM medium (Table 3). The experimental conditions were as follows: initial concentration of inoculum of 200 mg/L, temperature of 25 °C, aeration of 0.03 VVM (volume of air per volume of culture per minute), a photon flux density of 300 μmol/m^2^s, a residence time of 288 h, and a photoperiod of 16 h light and 8 h dark. Heterotrophic cultivation in dark conditions had an initial concentration of inoculum of 200 mg/L, constant agitation in orbital shaking plates at a speed of 120 rpm, a constant temperature of 25 °C and an initial pH of 7.5. In mixotrophic and heterotrophic culture, the initial concentration of organic carbon (glucose) added is 35 g/L.

### 3.5. Simulation of Microalgal Biomass Production and Techno-Economic Evaluation

In the second stage, six cases were considered for the cultivation stage and were evaluated with the help of the bioprocess simulator SuperPro Designer v10.0 (Table 4). The residence time was considered as the basis for the cases to be studied, because in the experiments, it was observed that the highest biomass and chlorophyll production is obtained at 72 h and 288 h of cultivation. For the simulation of this stage of the process, the preparation of the culture medium and the feeding of the airlift bioreactor were considered, using the data from the results of the experimental yields obtained in the first stage of the research. All simulation cases were evaluated in continuous operation mode, considering 330 days of annual operation [42]. To quantify the necessary bioreactors, it was considered that each one would cover an area of 4.6 m^2^ in order to make maximum use of sunlight in each one of them, and to avoid dark areas. Energy consumption by artificial light was not considered in the calculations, since it is considered that natural sunlight would be used. The total area to be covered by the cultivation stage was 1 ha, and each bioreactor has a maximum operating volume of 300 L.

The objective of the second stage was to evaluate which culture system would achieve the highest amount of biomass and the highest potential for chlorophyll production in a continuous process. Additionally, the energy consumption required to keep the bioreactors operating and the operating costs of the six cases were evaluated with the help of the simulator.

Operating costs include raw material, services and labor costs. The costs of services, labor and raw materials considered for this evaluation are presented in Table 3 and Table 5. Raw materials costs were obtained from data (updated as of 2023) from Marketplace (Alibaba.com, accessed on 2 February 2023); services and labor data were taken, as a reference, from the costs shown in the simulator.

For the third stage of the economic evaluation, we added to the simulation model of the cultivation stage (with the greatest potential for chlorophyll production), a proposal of the necessary stages for the extraction of chlorophyll produced by the consortium of cultivated microalgae. The proposed stages are shown in Figure 8; the operating conditions were taken from experimental results in the laboratory and some others from the literature.

Once the biomass is produced in the culture, it is necessary to move on to the primary harvesting stage. For primary harvesting, flocculation of the biomass with chitosan (1 g/L in solution with acetic acid and water 20:80 *v*/*v*) was considered, achieving a biomass concentration up to 50 g/L and recovering 80% of the biomass at this stage. Subsequently, the biomass is passed through a filter press, after which the biomass is additionally washed with water to eliminate impurities; this stage concentrates the biomass up to 200 g/L, and 90% of the biomass is recovered [43]. This technology was selected for this stage because it has been reported, compared to other technologies, to achieve the same objective (e.g., centrifugation or vacuum filtration), i.e., a technology with lower operating costs [43]. Finally, the collected biomass passes to the extraction stage, where only the use of solvents for this purpose was analyzed. To this end, the biomass is mixed with methanol in a 2:1 *v/v* ratio (biomass: solvent) for 5 min (mixer residence time), and then the mixture is kept at 4 °C for 24 h without turbulent agitation. Once the 24 h residence time has elapsed, the mixture is separated and the organic phase (methanol + chlorophyll) is passed to an evaporator, where the solvent is removed at 60 °C before being recirculated to the process (Figure 8). This last stage of the research aims to evaluate the impacts of the various factors that affect the cost of chlorophyll production using a microalgae consortium.

## 4. Conclusions

Experimentally, the mixotrophic culture (although it is not the one that achieves the highest biomass and chlorophyll concentration) simulation results obtained in a continuous operation mode show that the highest chlorophyll production would be achieved in this type of culture.

The process of extracting chlorophyll from a consortium of microalgae still has very high operating costs, and the greatest contribution to these costs is made by the high energy consumption of the closed photobioreactors used in the cultivation stage. For this stage, other types of cultivation systems with lower energy consumption should be considered, as should clean energy generation systems; additionally, cultivation conditions that favor an increase in biomass production should be sought. The second factor making a high contribution to the operating cost is the consumption of the flocculant agent for the primary harvesting of the biomass. This should therefore be evaluated in order to reduce the dosage and its effects on biomass harvesting, as well as its impact on operating costs. These two factors have the greatest impact on costs, and alternatives should be sought to reduce them, since the proposed process has great potential to be exploited.

Additionally, we should look for alternatives to increase the production of biomass and chlorophyll from microalgae consortia, considering the technology has already been established in the process design; this would also help to improve the economics of the process by combining the use of biomass with the extraction of other compounds of commercial interest in a biorefinery scheme. Additionally, we should consider recycling the water obtained in the primary harvest, and consider it as a product that can be marketed and therefore contribute to improving the economy of the process.

## Figures and Tables

**Figure 1 marinedrugs-21-00321-f001:**
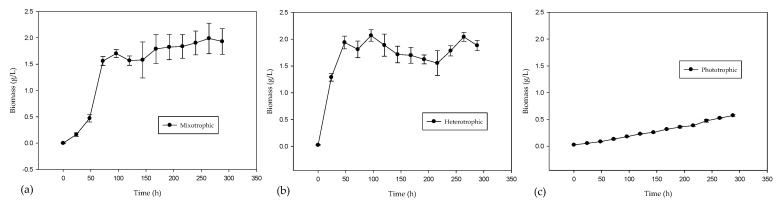
Biomass production in the different types of growth with 288 h of culture. (**a**) mixotrophic; (**b**) heterotrophic; (**c**) phototrophic (with error bars).

**Figure 2 marinedrugs-21-00321-f002:**
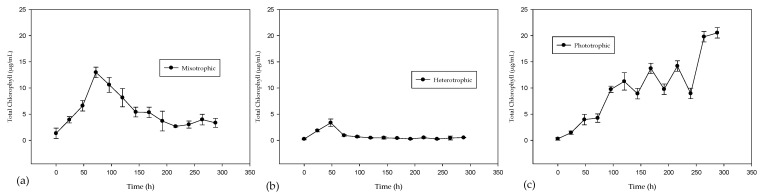
Total chlorophyll production in the different types of growth with 288 h of culture. (**a**) mixotrophic; (**b**) heterotrophic; (**c**) phototrophic (with error bars).

**Figure 3 marinedrugs-21-00321-f003:**
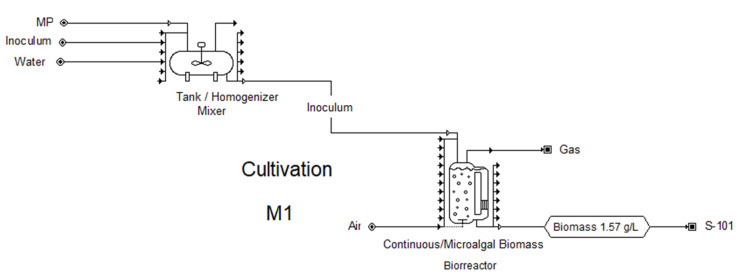
Flowsheet for the design of the microalgal biomass cultivation stage in a closed system, case M1.

**Figure 4 marinedrugs-21-00321-f004:**
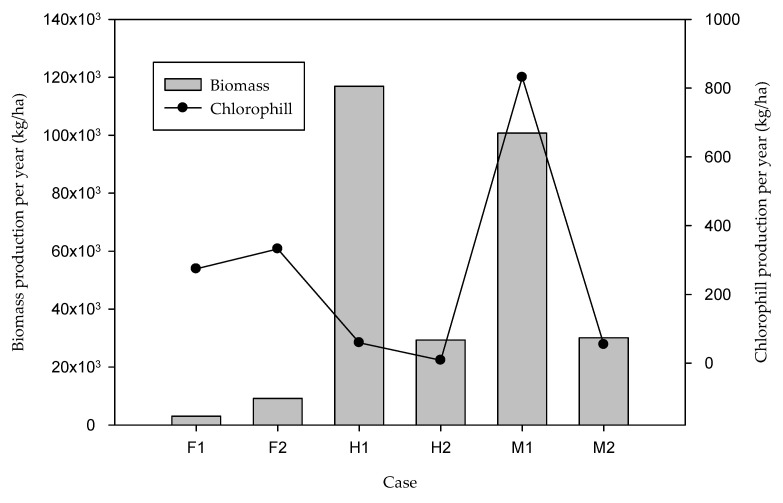
Biomass and chlorophyll production for all cultivation cases.

**Figure 5 marinedrugs-21-00321-f005:**
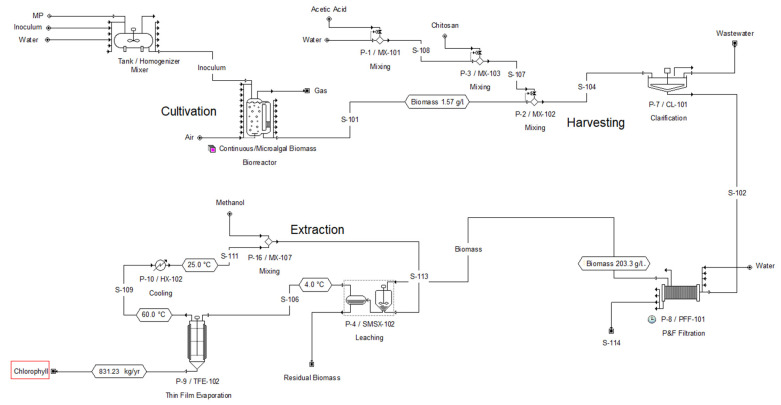
Flowsheet for the design of the chlorophyll extraction process from biomass of a microalgal consortium.

**Figure 6 marinedrugs-21-00321-f006:**
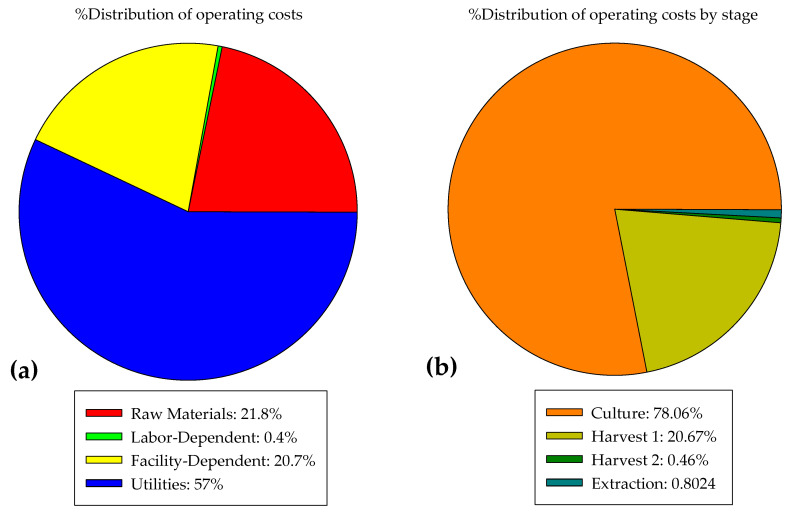
(**a**) Distribution of operating costs of the chlorophyll production process; (**b**) distribution of operating costs by stage of the process.

**Figure 7 marinedrugs-21-00321-f007:**
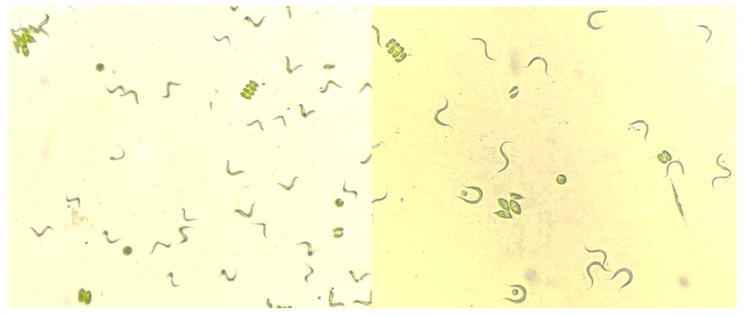
Morphology of the microalgae consortium.

**Figure 8 marinedrugs-21-00321-f008:**
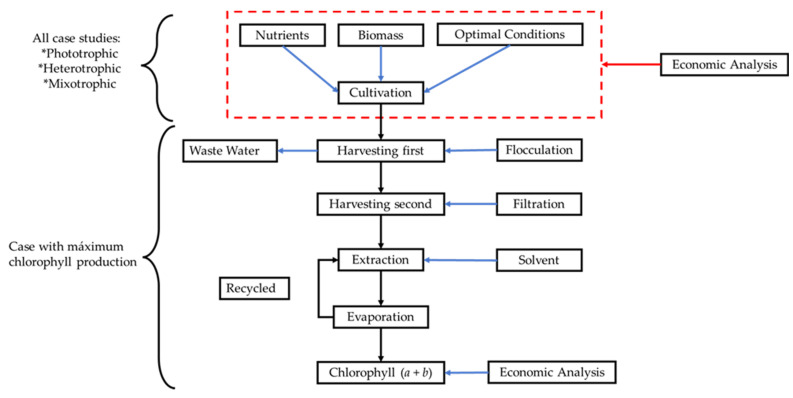
Simulation of the biomass and chlorophyll production process of the microalgae consortium.

**Table 1 marinedrugs-21-00321-t001:** Annual biomass production estimated with the help of the simulator.

Case	Biomass (g/L)	Chlorophyll (µg/L)
F1	0.04	4240
F2	0.56	20,540
H1	1.83	930
H2	1.81	520
M1	1.57	12,970
M2	1.85	3330

**Table 2 marinedrugs-21-00321-t002:** Equipment costs.

Equipment	Description	Units	Cost (USD)
Photobioreactor	Air lift fermenter (300 L)	2176	4,352,000
Homogenizer	Horizontal tank (10,000 L)	1	1000
Clarifier	CL-101	1	33,000
Plate and frame filter	PFF-101	1	72,000
Solid mixer–settler extractor	SMSX-102	1	11,000
Evaporator	TFE-102	1	60,000
Heat exchanger	HX-102	1	9000

**Table 3 marinedrugs-21-00321-t003:** BBM media composition [41].

Reagents	per Liter	Cost (USD/ton)
KH_2_PO_4_	175 mg	1650
CaCl_2_·2H_2_O	25 mg	120
MgSO_4_·7H_2_O	75 mg	95
NaNO_3_	250 mg	50
K_2_HPO_4_	75 mg	300
NaCl	25 mg	78
H_3_BO_3_	11.42 mg	300
*Microelements Stock Solution*		
ZnSO_4_·7H_2_O	8.82 g	2000
MnCl_2_·4H_2_O	1.44 g	500
MoO_3_	0.71 g	40,000
CuSO_4_·5H_2_O	1.57 g	13,000
Co(NO_3_)_2_·6H_2_O	0.49 g	9000
*Solution 1*		
Na_2_EDTA	50 g	1500
KOH	3.1 g	1300
*Solution 2*		
FeSO_4_	4.98 g	150
H_2_SO_4_ (Conc.)	1 mL	330

**Table 4 marinedrugs-21-00321-t004:** Description of case studies.

Case	Type of Crop	Residence Time (h)
F1	Phototrophic	72
F2	Phototrophic	288
H1	Heterotrophic	72
H2	Heterotrophic	288
M1	Mixotrophic	72
M2	Mixotrophic	288

**Table 5 marinedrugs-21-00321-t005:** Data of raw materials, services and labor costs.

Kind of Service	Price	Unity
Raw materials		
Acid acetic	730	USD/ton
Chitosan	224	USD/ton
Water	0.26	USD/m^3^
Services		
Std power	0.1	USD/kWh
Steam	12	USD/ton
Chilled water	0.4	USD/ton
Glycol	0.35	USD/ton
Labor		
Operator	0.37	USD/h

## Data Availability

The data that support the findings of this study are available from the corresponding author upon reasonable request.

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
