# Peer review of "Simulation and Economic Analysis of the Biotechnological Potential of Biomass Production from a Microalgal Consortium"

_marinedrugs, 2023, doi:10.3390/md21060321_

Round 1

Reviewer 1 Report

I find the manuscript "Simulation and Economic analysis of the Biotechnological potential of manuscript relates to the field of research on the biotechnological potential of biomass production from microalgal consortium and is currently relevant. It contains new interesting data concerning techno-economic evaluation for the production of chlorophyll and microalgal biomass, for the different types of growth in culture. Such studies could be very valuable considering the development of large-scale industrial cultivation of microalgae.  However, there are some weak points which should be considered:

General comments

The main results of the work include the following:

(1)  Experimental study of three types of microalgae biomass growth at the cultivation stage (phototrophic, heterotrophic and mixotrophic);

(2)  Simulation of microalgal biomass production and its techno-economic evaluation.

The main weak point of the work, in my opinion, is the almost complete absence of discussion of these two areas of results. The authors describe in detail the technical and economic assessment of the production of chlorophyll and microalgae biomass. At the same time, there is no detailed discussion of experimental results on changes in biomass and chlorophyll production in the different types of growth. The same remark applies to a theoretical model which was developed as a design proposal for the microalgal biomass production. Have other authors conducted similar studies? It is necessary to compare the original data with the data of other researchers and draw conclusions. Otherwise, we are dealing with a detailed economic and technological evaluation of cultivation processes performed by authors, and not with scientific work.

Furthermore, authors wrote that ‘Cultures with microalgae consortia are not very well studied because it is not known with certainty how the population interaction will react to the established conditions, therefore it is important to analyze all possible conditions’. (lines 131-134). Here it would be appropriate to analyze the results obtained with the data available in the scientific literature on the production of biomass and chlorophyll and on the conditions conducive to their maximum production in a microalgal consortium. In connection with that matter, some  relevant citations are  omitted, e.g. the following important and/or review papers concerning on the microalgae consortium are not cited:

Shahid, A., Malik, S., Zhu, H., Xu, J., Nawaz, M. Z., Nawaz, S.,  & Mehmood, M. A. (2020). Cultivating microalgae in wastewater for biomass production, pollutant removal, and atmospheric carbon mitigation; a review. Science of the Total Environment, 704, 135303.

Zhu, S., Huo, S., & Feng, P. (2019). Developing designer microalgal consortia: A suitable approach to sustainable wastewater treatment. Microalgae biotechnology for development of biofuel and wastewater treatment, 569-598.

Gonçalves, A. L., Pires, J. C., & Simões, M. (2017). A review on the use of microalgal consortia for wastewater treatment. Algal Research, 24, 403-415.

Minor comments.

1.     P. 11, lines 337-338. Authors wrote: ‘The proposed stages are shown in Figure 7, the operating conditions were taken from experimental results in the laboratory and some others from the 337 literature’.  Please specify exactly what conditions were indicated based on the literature data and what literature is meant.

2.     The numbering of the sections is incorrect. It is necessary to correct the numbering as the 4.Conclusions.

3.     Captions to Figures 1 and 2, it is better to make uniform: e.g. Biomass (total chlorophyll) production for different types of growth with a culture of 288 hours.

4.     There are unsuccessful phrases: e.g. ‘from it are of great interest for‘ (line 19), and ‘due to their interest in producing .. ‘ (lines 40-41).

Author Response

The document with the answers is attached

Reviewer 2 Report

Dear authors,

In this paper, the authors evaluate the utilization of a consortium of microalgae to increase biomass and chlorophyll production. I believe there are several deficiencies that lead me to recommend that the paper should not be accepted for publication until they are addressed.

Majors:

L30:” The highest concentration was 2.06 g/L in heterotrophic culture, followed 30by (1.98 g/L)” The highest concentration of what? Please define

*The introduction is well written, I just miss an important point, which is to describe that microalgae are also being used in consortia with bacteria for different types of biotechnological applications

*The legend of Figure 1 and 2 is insufficient to understand it. The figures do not present statistical errors, error bars, or the number of repetitions performed.

L272: “Study material” How have the bacteria present in the collected samples been eliminated? This is a key point that needs to be clarified. Are you sure that there is no bacterial contamination? Why?

L272: “Study material” I don't think it is appropriate to identify the algae in the samples only by observation. It is not rigorous. There was also no quantification of the relative concentration. They should show at least photos indicating the presence of these algae and the absence of bacteria or other organisms.

L112: “can be attributed to the ability of the consortium to take advantage of glucose as a carbon source” Which of the algae present in the samples can use glucose as a carbon source? Please indicate and discuss it.

*Why, at time zero in Figures 1 and 2, do the three points not coincide? I don't understand it. Could you please explain it.

* Resolving whether there is bacterial contamination is crucial because that's what it seems by analyzing the data in Figures 1 and 2. In Figure 2, chlorophyll in mixotrophic and heterotrophic cultures decreases over time, whereas in Figure 1, biomass in such cultures increases. I am afraid that a good portion of the biomass they are measuring may be bacterial contamination.

*Figure 4 also has no statistical errors. The figure legend is also insufficient. At the very least, it should clearly indicate what F1, F2, etc. mean.

*The meaning of Figure 3 is not really explained. Where does the value of 1.57 g/L come from.

* No references are provided to support where any of the costs indicated come from.

* The meaning of Figure 5 is impossible to understand with the explanations provided by the authors

L166-L179: There is no bibliographic support presented to justify such economic data

Author Response

(The authors gave the same response as above.)

Reviewer 3 Report

The manuscript Is original and relevant

Materials que Methods well used

In my opinion, the authors should expand the discussion since they say it Is an expensive method for Cultures. Please view More examples

From other country use tu method.

Improve some grammatical forms.

Author Response

(The authors gave the same response as above.)

Round 2

Reviewer 1 Report

The authors have made the necessary corrections according to the comments I made earlier. I recommend accept the manuscript in its present form.

Reviewer 2 Report

Dear authors

I believe the authors have responded positively to most of my comments, and I accept the paper with those changes